# Comparison of Risk Factors for Cholangiocarcinoma and Hepatocellular Carcinoma: A Prospective Cohort Study in Korean Adults

**DOI:** 10.3390/cancers14071709

**Published:** 2022-03-28

**Authors:** In Rae Cho, Sang-Wook Yi, Ja Sung Choi, Jee-Jeon Yi

**Affiliations:** 1Department of Internal Medicine and Liver Research Institute, Seoul National University College of Medicine, 101 Daehak-ro, Jongno-gu, Seoul 03080, Korea; inrae0428@hanmail.net; 2Department of Preventive Medicine and Public Health, College of Medicine, Catholic Kwandong University, Bumil-ro 579 beon-gil 24, Gangneung 25601, Korea; 3Department of Internal Medicine, International St. Mary’s Hospital, Catholic Kwandong University College of Medicine, 25 Simgok-ro 100 beon-gil, Seo-gu, Incheon 22711, Korea; cjs0123@hanmail.net; 4Institute for Occupational and Environmental Health, Catholic Kwandong University, Bumil-ro 579 beon-gil 24, Gangneung 25601, Korea; yihaeyou@gmail.com

**Keywords:** cholangiocarcinoma, hepatocellular carcinoma, risk factor, liver cirrhosis

## Abstract

**Simple Summary:**

Cholangiocarcinoma (CCA), especially intrahepatic cholangiocarcinoma, shares many of the commonly cited risk factors for hepatocellular carcinoma (HCC). Therefore, a common pathogenesis has been suggested for HCC and CCA; liver cirrhosis (LC) is considered a key factor in this “common pathway” hypothesis. In this large-scale prospective cohort study in Koreans, choledocholithiasis, cholelithiasis, HBV infection, older age, male sex, diabetes, smoking, alcohol drinking, and obesity were found to be potential risk factors for CCA. In the current study, LC (the most important risk factor of HCC) did not increase the risk for CCA and there were also differences between CCA and HCC in the magnitude of common risk factors. Our study suggests that there is weak epidemiologic evidence for the hypothesis that LC is a key common factor involved in the pathogenesis of both CCA and HCC.

**Abstract:**

Cholangiocarcinoma (CCA), especially intrahepatic CCA, is known to share several risk factors with hepatocellular carcinoma (HCC) and liver cirrhosis has been proposed as a common pathogenic factor. We aimed to identify the risk factors of CCA and to examine differences in risk factors between CCA and HCC. We followed 510,217 Korean adults who underwent health checkups during 2002–2003 until 2013 via linkage to national hospital discharge records. Hazard ratios (HRs) were calculated after adjustment for confounders. During the mean follow-up of 10.5 years, 1388 and 2920 individuals were diagnosed with CCA and HCC, respectively. Choledocholithiasis (HR = 13.7; 95% confidence interval (CI) = 7.58–24.88) was the strongest risk factor for CCA, followed by cholelithiasis (HR = 2.94) and hepatitis B virus (HBV) infection (HR = 2.71). Two of the strongest risk factors for HCC—liver cirrhosis (HR = 1.29; 95% CI = 0.48–3.45) and hepatitis C virus infection (HR = 1.89; 95% CI = 0.49–7.63)—were not significantly associated with the risk of CCA. HBV infection and diabetes increased the risk of both HCC and CCA, but the HRs were lower for CCA than for HCC (*P*_heterogeneity_ < 0.001 for HBV; *P*_heterogeneity_ = 0.001 for diabetes). The magnitudes of the effects of age, sex, obesity, alcohol consumption, and smoking on the development of both cancers were different (*P*_heterogeneity_ < 0.05 for each variable). In conclusion, choledocholithiasis, cholelithiasis, HBV, older age, male sex, diabetes, smoking, alcohol drinking, and obesity were found to be potential risk factors of CCA. Liver cirrhosis did not increase the risk of CCA. The magnitudes of the potential effects of common risk factors were generally different between CCA and HCC.

## 1. Introduction

Both cholangiocarcinoma (CCA) and hepatocellular carcinoma (HCC) exhibit aggressive features and their incidence has increased worldwide over the past decades [1,2,3]. In 2021, an estimated 23,000 new CCA and HCC cases were diagnosed in Korea and 15,000 related deaths occurred [4]. Therefore, identifying their risk factors and high-risk individuals is important for prevention and early diagnosis [5]. CCA, especially intrahepatic cholangiocarcinoma (iCCA), is known to share many of the commonly cited risk factors for HCC [5]. This fact has been interpreted as suggesting a common pathogenesis for these two primary liver cancers: HCC and iCCA [6]. Among the relevant risk factors, liver cirrhosis (LC) is considered a key factor in this “common pathway” hypothesis, since the relative risk of LC for HCC and iCCA development has commonly been reported to be ≥10 and most of the common risk factors for both CCA and HCC (e.g., viral hepatitis and alcohol consumption) are associated with LC [7,8,9].

However, compared with HCC, fewer studies have identified the risk factors of CCA in a prospective manner in the general population with adjustments for demographic, behavioral, metabolic, and medical risk factors [10]. Some risk factors of CCA, especially medical risk factors such as LC and viral hepatitis, have mainly been derived from case-control studies [11,12], in which selection bias and information bias regarding risk factors are more likely to occur than in prospective cohort studies.

Therefore, we aimed to identify the risk factors of CCA through a large-scale prospective cohort study [13]. Additionally, we tried to examine epidemiologic evidence for the hypothesis that HCC and CCA share a common pathogenesis, especially through LC, by comparing the risk factors between CCA and HCC. Furthermore, we examined potential differences in risk factors between iCCA and extrahepatic cholangiocarcinoma (eCCA).

## 2. Patients and Methods

### 2.1. Study Population and Follow-Up

The National Health Insurance Service (NHIS) provides mandatory healthcare coverage to 97% of the Korean population [14]. The study cohort (*n* = 514,866) comprised a random sample including 10% of the 5.15 million NHIS beneficiaries between 40 and 79 years of age in 2002 who underwent a health examination between 2002 and 2003. From this sample, 2887 persons with a history of cancer were excluded, as were 1762 subjects with missing information. The remaining 510,217 individuals were followed until 31 December 2013 through linkage to the NHIS hospital discharge records, which are generated by certified health information managers who assign standardized diagnosis codes after reviewing patients’ medical records (Figure 1). The cancer incidence data of the NHIS is complete enough to be compared with the data of the Korea National Cancer Incidence Database and a previous analysis estimated the concordance as 97.8% [15,16]. Patients discharged from the hospital with the following International Statistical Classification of Diseases and Related Health Problems, 10th revision (ICD-10) codes were identified as HCC and CCA cases: CCA (C221, C240, C248, and C249), iCCA (C221), extrahepatic CCA (eCCA: C240, C248, and C249) and HCC (C220) [13]. This study was approved by the Institutional Review Board of Catholic Kwandong University, Republic of Korea (approval number: CKU-16-01-0301). The requirement for informed consent was waived because this study analyzed anonymized data that were provided by the NHIS according to strict confidentiality guidelines.

### 2.2. Data Collection

Direct measurements and a questionnaire were used to collect data at the baseline health examinations in 2002–2003. The Reitman–Frankel method or the nicotinamide adenine dinucleotide ultraviolet absorption method was used to measure aspartate transaminase (AST) and alanine transaminase (ALT) levels. Total cholesterol (TC) and serum glucose levels were measured using fasting serum samples. Body mass index (BMI) was calculated using measured values of weight and height (kg/m^2^). A questionnaire was used to assess patients’ history of cancer, alcohol use, and smoking status.

The amount of alcohol consumption was estimated according to the method used in our previous study [13]. Participants were classified by their amount of alcohol consumption (g ethanol/day) into 4 groups (none, <10, 10–39, ≥40, and missing information). Alcohol consumption measures were further validated by analyzing associations with the incidence of alcoholic liver disease during follow-up.

### 2.3. Medical Risk Factors at Baseline

Individuals were considered to have a pre-existing disease if they had made at least one visit to a medical institution for a diagnosed disease in the interval extending from 6 months before the baseline health examination date to 2 months after. The medical risk factors for CCA and HCC were identified using the following ICD-10 codes: viral hepatitis, B15–B19; hepatitis B virus (HBV) infection, B16, B180, B181; hepatitis C virus (HCV) infection, B171, B182; diabetes, E10–E14; non-viral liver disease, K70–K76; ALD, K70; liver cirrhosis, K74; non-alcoholic fatty liver disease (NAFLD) and non-alcoholic steatohepatitis (NASH), K758 and K760; cholelithiasis, K80; gallbladder stone, K800–K802; and bile duct stone, K803–K805.

### 2.4. Statistical Analysis

Participants were classified by smoking status as never, former, or current smokers [<1 pack/day or ≥1 pack/day], and missing information; their diabetes status was classified as normoglycemia (fasting glucose <100 mg/dL), impaired fasting glucose (100–125 mg/dL), and diabetes (≥126 mg/dL or a previous diagnosis of diabetes). Additionally, to investigate the possibility of a dose–risk relation in the range of 90–299 mg/dL, fasting glucose levels were analyzed as a continuous variable (per 18 mg/dL (1.0 mmol/L) increase). BMI was classified as <18.5, 18.5–24.9, 25–29.9, and ≥30 kg/m^2^ and was also analyzed as a continuous variable (per 5 kg/m^2^ increase). TC was analyzed as a continuous variable (per 39 mg/dL (1 mmol/L) increase) [17].

The hazard ratios (HRs) and 95% confidence intervals (Cis) for CCA (including iCCA and eCCA) and HCC incidence were calculated using Cox proportional hazards models. In the multivariable analysis, adjustments were made for age at baseline (continuous variable per 10-year increment in age), sex, alcohol use, smoking status, BMI, diabetes status, physical activity (exercise that induces sweating at least once per week), income status (income deciles; 1st to 3rd [low income], 4th to 7th, 8th or more (high income)), and HBV infection. In patients with HCC, an additional adjustment was done for HCV infection and LC and an adjustment for choledocholithiasis (K803–K805) was performed for CCA. Cochran’s Q statistic was used as a heterogeneity test to examine differences in the effect size of potential risk factors between the two cancers.

Subanalyses were performed with stratification by age, sex, and the presence of a liver disorder. A liver disorder was considered to present in participants who had an AST level ≥ 40 IU/L, an ALT level ≥ 40 IU/L, non-viral liver disease, or viral hepatitis, while their counterparts (i.e., participants with an ALT level < 40 IU/L, an AST level < 40 IU/L, and no known liver disease) were considered not to have a liver disorder. All *p*-values were 2-sided. All analyses were conducted using SAS version 9.4 (SAS Institute Inc., Cary, NC, USA).

## 3. Results

### 3.1. Baseline Characteristics

The baseline characteristics of the study population are summarized in Table 1. Of the participants, their mean age was 50.3 ± 9.7 years and 277,067 (54.3%) were men. During the mean follow-up of 10.5 years, 2920 and 1388 individuals were diagnosed with HCC and CCA (821 with iCCA and 567 with eCCA), respectively. The median time intervals between the health examination and diagnosis of HCC and CCA were 5.3 years (interquartile range: 2.5–8.1) and 6.1 years (3.0–8.6), respectively. Patients with both cancers were older than the non-patients and showed a higher proportion of participants with male sex, current smoking, heavy alcohol consumption, low-income status, diabetes, viral hepatitis, and cholelithiasis.

### 3.2. Identification and Comparison of Risk Factors of CCA and HCC

Choledocholithiasis was the strongest risk factor for CCA (HR = 13.69 in the sex- and age-adjusted analysis; 13.73 in the multivariable analysis), followed by cholelithiasis and HBV infection (Table 2, Appendix A). Older age, male sex, smoking, alcohol consumption, obesity, and diabetes were also associated with the development of CCA. In HCC patients, LC (HR = 44.50 in the sex- and age-adjusted analysis; 19.38 in the multivariable analysis), HBV infection (HR = 22.03 in the sex- and age-adjusted analysis; 11.80 in the multivariable analysis), and HCV infection (HR = 18.38 in the sex- and age-adjusted analysis; 7.99 in the multivariable analysis) were strong risk factors. Older age, male sex, smoking, alcohol consumption, diabetes, elevated ALT, and lower cholesterol levels were also associated with the development of HCC.

When comparing the HRs of risk factors between CCA and HCC, the effect of aging was stronger in CCA development (2.59 vs. 1.62; *p* < 0.001), while differences by sex were more remarkable for the development of HCC (1.91 vs. 3.77; *p* < 0.001). Moderate alcohol consumption (10–39 g ethanol/day) and heavy smoking (≥1 pack/day) were significantly related with CCA development, but not with HCC. Liver cirrhosis and HCV infection, which were identified as risk factors of HCC, did not significantly increase the risk of CCA development. HBV infection and diabetes were associated with an increased risk of both HCC and CCA, but the HRs were significantly higher for HCC. HCC risk increased with higher ALT levels and lower TC, but this trend was relatively weak for CCA. In contrast, choledocholithiasis had a significantly higher HR for CCA than for HCC.

### 3.3. Differences in Risk Factors between iCCA and eCCA

The risk factors of iCCA and eCCA and the differences between the two subtypes are presented in Table 3. In summary, none of the risk factors that contributed to CCA development showed a statistically significant difference between the two subtypes of CCA, except for TC. Diseases related to chronic inflammation of the liver parenchyma, such as LC and viral infection, showed a non-significantly higher HR for iCCA than for eCCA and the HRs of diabetes and choledocholithiasis were non-significantly higher for eCCA.

### 3.4. Analysis According to the Presence of a Liver Disorder

In the stratified analysis according to the presence of a liver disorder, alcohol consumption showed different effects on the development of the two cancers (Table 4). The carcinogenic effect of frequent alcohol consumption (almost daily) on HCC development disappeared in the liver disorder group (1.27 vs. 0.91; *p* = 0.046), whereas frequent and heavy alcohol consumption was a significant risk factor for CCA development, regardless of the presence of a liver disorder.

## 4. Discussion

In the present cohort study in Koreans, choledocholithiasis, cholelithiasis, HBV infection, older age, male sex, diabetes, smoking, alcohol drinking, and obesity were associated with increased risk of CCA. When comparing the risk factors of CCA and HCC, the most remarkable finding was that LC, the strongest risk factor of HCC, did not significantly increase the risk of CCA development. In addition, statistical tests showed that the magnitude of risk associated with the common risk factors (e.g., age, sex, alcohol, obesity, and hepatitis B virus infection) differed between CCA and HCC.

In previous case-control studies, LC has been identified as the strongest risk factor for CCA, except for biliary tract diseases; this has been considered to be the epidemiologic basis of the “common pathway” hypothesis for CCA and HCC [6,11,18,19]. However, in this prospective cohort study, LC did not show a significant association with the development of CCA. The strong association between LC and CCA in previous studies, which were mostly case-control studies, may have been a consequence of their retrospective design. For example, in patients with CCA with symptoms such as abdominal pain or jaundice, subclinical liver disease may be found during the work-up. Similarly, in our study cohort, the multivariable-adjusted HR of LC was 4.4 (*p* < 0.001) when LC was defined as a diagnosis within two years before CCA diagnosis or the end of follow-up, instead of at baseline.

Other evidence exists that is generally in accordance with our findings. In large-scale nested case-control and cohort studies conducted in Asia, LC showed relatively low odds ratios for CCA (1.7 and 2.0) in persons with diabetes and was not found to be a significant risk factor for CCA (HR 1.06; 95% CI 0.87–1.29) in persons with biliary tract disease [20,21]. Differences in the etiology of LC might play a role in this discrepancy. In Swedish and Danish cohort studies, no CCA cases were found in persons with cirrhosis originating from viral hepatitis, whereas alcoholic cirrhosis was strongly associated with CCA, although no adjustment for modifiable risk factors was made [22,23]. The majority of LC cases are caused by HBV in Korea, whereas other etiologies such as alcohol, HCV, diabetes, and NAFLD are commonly implicated in LC development in Western countries [24,25,26,27].

In our study, HBV was associated with a three-fold increased risk of CCA, while HCV was associated with a non-significantly higher risk of CCA. These results are somewhat similar to those of previous studies in Eastern Asian nations, in which the prevalence of HBV infection was higher than that of HCV infection [21,28,29,30]. In the evidence from previous studies, it was unclear whether CCA development is a direct consequence of the virus itself or is mediated through LC that develops due to a viral infection. The specific characteristics of HBV-associated iCCA [5,31], combined with our finding that LC was not strongly associated with CCA development, suggest that the virus itself, rather than the eventual development of LC from a viral infection, is related to CCA risk. Additionally, it is worth noting that our estimated relative risks of viral hepatitis for CCA are lower than those reported for CCA in previous retrospective studies [6,12], as well as those for HCC in the current study.

Smoking and alcohol drinking were strongly associated with both CCA and HCC, as has been found in previous studies [12]; however, the pattern of the associations was somewhat different between CCA and HCC. The risk of CCA development increased proportionally to the amount of smoking, but the dose–risk relationship was less clear for HCC development, since participants who smoked ≥20 cigarettes/day had lower HRs than those who consumed <20 cigarettes/day. Both moderate (10–39 g/day) and heavy (≥40 g/day) alcohol use were associated with CCA development, whereas only heavy alcohol use was associated with HCC. Additionally, CCA showed a significant dose–risk relationship with alcohol, regardless of the presence of liver disease. In contrast, the effect of alcohol on HCC development disappeared in persons with underlying liver diseases. These results suggest that liver disease plays a role as a mediator between alcohol drinking and HCC development, but not in CCA development.

Other common risk factors also had different magnitudes of relative risk for HCC and CCA development. Increasing age had a greater impact on CCA than on HCC, while HCC showed a stronger pattern of sexual dimorphism (higher risk in men than in women) [32,33]. Obesity showed stronger associations with CCA than with HCC. Obesity may increase the risk of CCA directly and it may also indirectly increase CCA risk through gallstone formation, which is a strong risk factor for CCA [34,35,36,37]. Diabetes showed a stronger effect on HCC development than on CCA development, in accordance with previous reviews [38,39,40,41,42]. The stronger impact of diabetes on HCC development than on CCA may reflect the fact that diabetes and chronic liver disease can affect each other [43,44,45]. Higher ALT levels and lower TC levels, which are markers of liver disease and its severity, were strongly associated with the risk of HCC, but they had a weak relationship with CCA.

The strongest risk factors for CCA were choledocholithiasis and cholelithiasis, as in a previous study [11]. In this study, no significant difference was found in the HRs of potential risk factors between the two subtypes of CCA, except that high ALT levels were associated with a higher risk of iCCA but not eCCA and that TC had a stronger inverse association with iCCA than with eCCA. These results suggest that liver damage may have a stronger relationship with iCCA, a type of primary liver cancer, than eCCA, since high ALT and lower TC are markers of liver damage. The existing literature comparing risk factors between iCCA and eCCA in a large-scale and prospective manner is scarce. Although some case-control studies have reported differences in risk factors between these two subtypes of CCA, no risk factors have shown consistent results among studies [11,19,20,46]. It remains unclear whether there is a difference in risk factors between iCCA and eCCA, perhaps except for factors directly related to liver damage, which may have stronger effects on iCCA than on eCCA.

According to the common pathway hypothesis for the development of CCA and HCC, hepatic progenitor cells can differentiate into both cancer types [6,47]. LC, which is regarded as the final common pathological pathway of liver damage arising from a wide variety of chronic liver diseases, is considered to be a key component of this “common pathway”; however, in the current study LC was not associated with an increased risk of CCA. As discussed above, most of the common risk factors of HCC and CCA showed different effect sizes for these two carcinomas. CCA (or more specifically iCCA) may be less strongly affected by chronic liver inflammation than HCC and the deleterious consequences of viral infections and alcohol may have direct effects on CCA, rather than exerting indirect effects through a chronic liver disease mediated mechanism. Therefore, our study suggests that there may be differences in the pathogenesis between HCC and CCA, even if hepatic progenitor cells differentiate into HCC and CCA.

This study has several strengths, the first of which is its prospective nature, which minimized recall and selection bias related to a retrospective design. Second, differences in the effect size of potential risk factors between cancers in the same study population were directly compared using formal statistical tests. Additionally, the large number of cancer cases with nearly complete follow-up enabled the examination of the most important risk factors including behavioral, metabolic, and medical factors [13]. However, there are also limitations. First, the fact that this study population comprised Koreans may affect the generalizability of the findings, especially to other ethnic populations with a different distribution of LC etiologies. Second, risk factors were established from single measurements, meaning that the relative risks of potential risk factors might have been somewhat underestimated due to the regression dilution effect. Third, it was difficult to analyze diseases with low incidence or congenital anomalies, therefore some infrequent risk factors were not assessed. Fourth, the time interval from the health examination to the cancer diagnosis was relatively short for some cases; for example, 315 CCA patients and 852 HCC patients were diagnosed within three years after the health examination. Therefore, the impact of some factors might reflect the early presentation of CCA and HCC to some degree. However, excluding the first three years of follow-up led to no material changes in the associations compared with the main analysis. Additionally, excluding individuals with previous history of any cancer might have led to bias.

## 5. Conclusions

Through this prospective cohort study in Koreans, choledocholithiasis, cholelithiasis, HBV infection, older age, male sex, diabetes, smoking, alcohol drinking, and obesity were found to be potential risk factors of CCA. However, LC, the most important risk factor of HCC, did not increase the risk for CCA and there were also differences between CCA and HCC in the magnitude of common risk factors. There were generally no differences in potential risk factors between iCCA and eCCA. Our study suggests that the epidemiologic evidence for the hypothesis that LC is a key factor involved in the common carcinogenesis of both CCA and HCC is weak.

## Figures and Tables

**Figure 1 cancers-14-01709-f001:**
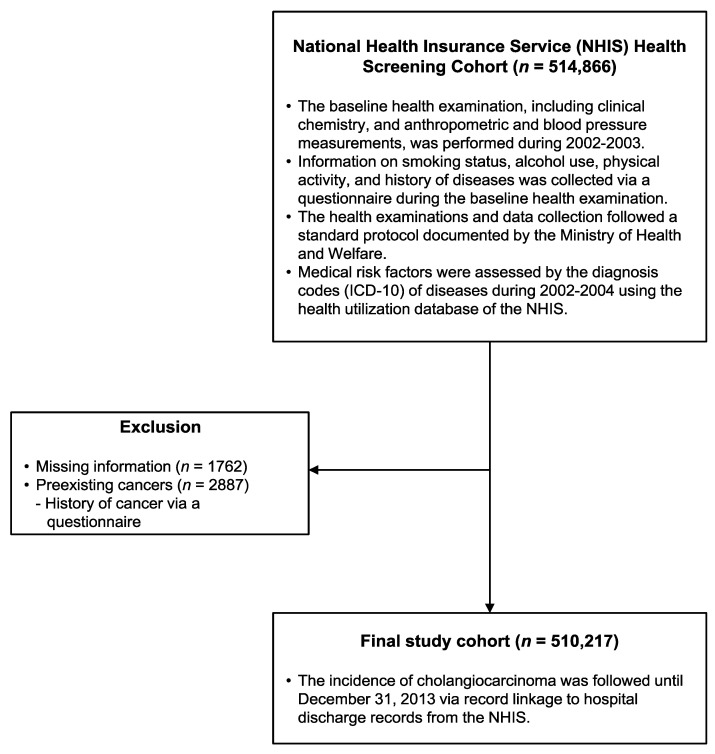
Flow diagram of the study population and design.

**Table 1 cancers-14-01709-t001:** Baseline characteristics of study population.

Variable/Group	Total Cohort		CCA Cases			iCCA Cases			eCCA Cases			HCC Cases		
	*n* = 510,217		*n* = 1388		*p*	*n* = 821		*p*	*n* = 567		*p*	*n* = 2920		*p*
Age, years	53.0	±9.7	62.0	±9.1	<0.001	61.3	±9.2	<0.001	63.0	±8.9	<0.001	56.2	±9.4	<0.001
Body mass index, kg/m^2^	24.0	±3.0	24.1	±3.0	0.444	24.0	±3.0	0.751	24.1	±3.1	0.415	23.9	±3.0	0.236
Fasting glucose, mg/dL	98.4	±34.8	104.8	±43.5	<0.001	103.6	±42.2	<0.001	106.4	±45.3	<0.001	108.2	±49.3	<0.001
Total cholesterol, mg/dL	200.5	±38.7	197.6	±38.6	0.006	195.4	±37.7	<0.001	200.9	±38.7	0.811	177.9	±37.2	<0.001
ALT, IU/L	26.1	±21.6	28.9	±25.2	<0.001	29.1	±22.7	<0.001	28.6	±28.5	<0.001	56.5	±54.9	<0.001
AST, IU/L	27.1	±18.4	31.4	±19.9	<0.001	31.7	±20.2	<0.001	30.9	±19.5	<0.001	60.5	±50.4	<0.001
Sex, men	277,067	(54.3)	945	(68.1)	<0.001	565	(68.8)	<0.001	363	(64.0)	<0.001	2407	(82.4)	<0.001
*Smoking status*					<0.001			<0.001			0.05			<0.001
Never smoker	327,361	(64.2)	794	(57.2)		459	(55.9)		328	(57.8)		1424	(48.8)	
Past smoker	43,311	(8.5)	143	(10.3)		82	(10.0)		55	(9.7)		347	(11.9)	
Current smoker, <1 pack/day	92,113	(18.1)	318	(22.9)		191	(23.3)		119	(21.0)		863	(29.6)	
≥1 pack/day	25,868	(5.1)	84	(6.1)		57	(6.9)		27	(4.8)		158	(5.4)	
Missing data	21,564	(4.2)	49	(3.5)		32	(3.9)		16	(2.8)		128	(4.4)	
*Alcohol use, g ethanol/day*					<0.001			<0.001			<0.001			<0.001
None	282,941	(55.5)	743	(53.5)		433	(52.7)		299	(52.7)		1387	(47.5)	
<10	101,672	(19.9)	222	(16.0)		125	(15.2)		94	(16.6)		542	(18.6)	
10–39	89,512	(17.5)	277	(20.0)		172	(21.0)		99	(17.5)		632	(21.6)	
≥40	24,260	(4.8)	120	(8.6)		70	(8.5)		48	(8.5)		284	(9.7)	
Missing data	11,832	(2.3)	26	(1.9)		21	(2.6)		5	(0.9)		75	(2.6)	
*Physical activity*					0.001			0.003			0.118			0.701
≥1 times/week	209,650	(41.1)	512	(36.9)		296	(36.1)		206	(36.3)		1210	(41.4)	
*Income status, decile*					<0.001			0.002			0.001		(0.0)	<0.001
<4 (low-income)	117,458	(23.0)	378	(27.2)		223	(27.2)		151	(26.6)		738	(25.3)	
4–7	166,212	(32.6)	480	(34.6)		278	(33.9)		193	(34.0)		993	(34.0)	
>7 (high-income)	226,547	(44.4)	530	(38.2)		320	(39.0)		201	(35.4)		1189	(40.7)	
*Body mass index,* kg/m^2^					0.435			0.822			0.372			0.794
<18.5	11,722	(2.3)	35	(2.5)		18	(2.2)		16	(2.8)		73	(2.5)	
18.5–24.9	319,295	(62.6)	847	(61.0)		504	(61.4)		326	(57.5)		1840	(63.0)	
25–29.9	164,528	(32.2)	458	(33.0)		272	(33.1)		183	(32.3)		927	(31.7)	
≥30	14,672	(2.9)	48	(3.5)		27	(3.3)		20	(3.5)		80	(2.7)	
*Diabetes status*					<0.001			<0.001			<0.001			<0.001
Normoglycemia	342,697	(67.2)	796	(57.3)		493	(60.0)		292	(51.5)		1640	(56.2)	
IFG	114,658	(22.5)	341	(24.6)		192	(23.4)		145	(25.6)		693	(23.7)	
Diabetes	52,862	(10.4)	251	(18.1)		136	(16.6)		108	(19.0)		587	(20.1)	
*ALT, IU/L*					<0.001			<0.001			<0.001			<0.001
<20	220,083	(43.1)	481	(34.7)		289	(35.2)		192	(33.9)		299	(10.2)	
20–39	224,450	(44.0)	691	(49.8)		394	(48.0)		297	(52.4)		1029	(35.2)	
40–59	43,744	(8.6)	139	(10.0)		88	(10.7)		51	(9.0)		711	(24.3)	
60–79	12,065	(2.4)	43	(3.1)		28	(3.4)		15	(2.6)		372	(12.7)	
≥80	9875	(1.9)	34	(2.4)		22	(2.7)		12	(2.1)		509	(17.4)	
*Comorbid liver disease*														
Viral hepatitis	4655	(0.9)	25	(1.8)	<0.001	18	(2.2)	<0.001	7	(1.2)	0.361	401	(13.7)	<0.001
Hepatitis B virus infection	2712	(0.5)	14	(1.0)	0.014	9	(1.1)	0.026	5	(0.9)	0.215	287	(9.8)	<0.001
Hepatitis C virus infection	516	(0.1)	3	(0.2)	0.177	2	(0.2)	0.199	1	(0.2)	0.545	57	(2.0)	<0.001
Liver flukes	361	(0.1)	1	(0.1)	0.986	1	(0.1)	0.582	0	(0.0)	0.534	1	(0.0)	0.457
Nonviral liver disease	14,334	(2.8)	60	(4.3)	0.001	39	(4.8)	0.001	19	(3.4)	0.339	548	(18.8)	<0.001
Liver cirrhosis	1114	(0.2)	4	(0.3)	0.577	3	(0.4)	0.366	1	(0.2)	0.862	255	(8.7)	<0.001
Alcoholic liver disease	2864	(0.6)	15	(1.1)	0.01	7	(0.9)	0.264	8	(1.4)	0.005	81	(2.8)	<0.001
NASH/NAFLD	2010	(0.4)	7	(0.5)	0.511	6	(0.7)	0.123	1	(0.2)	0.433	16	(0.5)	0.183
*Biliary tract disease*														
Cholelithiasis	878	(0.2)	18	(1.3)	<0.001	9	(1.1)	<0.001	8	(1.4)	<0.001	13	(0.4)	<0.001
Choledocholithiasis	212	(0.0)	11	(0.8)	<0.001	5	(0.6)	<0.001	5	(0.9)	<0.001	2	(0.1)	0.474

Abbreviations: CCA—cholangiocarcinoma; iCCA—intrahepatic cholangiocarcinoma; eCCA—extrahepatic cholangiocarcinoma; HCC—hepatocellular carcinoma; IFG—impaired fasting glucose; ALT—alanine aminotransferase; NASH—nonalcoholic steatohepatitis; NAFLD—nonalcoholic fatty liver disease. Data were expressed as the mean ± standard deviation or *n* (%).

**Table 2 cancers-14-01709-t002:** Multivariable-adjusted HRs of CCA and HCC.

Variable/Group		CCA Incidence ^a^			HCC Incidence ^b^				*p*-Value for Heterogeneity
	No. of Cases	*p*	HR	(95% CI)	No. of Cases	*p*	HR	(95% CI)	
Age, years									
Per 10-year older	1388	<0.001	2.59	(2.13–3.14)	2920	<0.001	1.62	(1.42–1.85)	<0.001
Sex, men (vs. women)	945	<0.001	1.91	(1.66–2.19)	2407	<0.001	3.77	(3.38–4.20)	<0.001
*Smoking status*									
Never smoker	794		1.00	(Reference)	1424		1.00	(Reference)	
Past smoker	143	0.154	1.15	(0.95–1.36)	347	0.163	1.09	(0.97–1.24)	0.664
Current smoker, <1 pack/day	318	<0.001	1.31	(1.13–1.52)	863	<0.001	1.35	(1.23–1.49)	0.701
≥1 pack/day	84	0.002	1.46	(1.14–1.85)	158	0.116	0.87	(0.73–1.03)	0.001
*Alcohol use, g ethanol/day*									
None	743		1.00	(Reference)	1387		1.00	(Reference)	
<10	222	0.257	0.91	(0.78–1.07)	542	0.002	0.84	(0.76–0.94)	0.430
10–39	277	0.032	1.19	(1.01–1.39)	632	0.402	0.96	(0.86–1.06)	0.025
≥40	120	<0.001	1.54	(1.25–1.89)	284	<0.001	1.40	(1.22–1.60)	0.460
Per 140 g ethanol/week increase	1388	<0.001	1.12	(1.06–1.17)	2920	<0.001	1.11	(1.07–1.15)	0.869
*Body mass index, kg/m* * ^2^ *									
<18.5	35	0.173	0.79	(0.56–1.11)	73	0.975	1.00	(0.79–1.26)	0.270
18.5–24.9	847		1.00	(Reference)	1840		1.00	(Reference)	
25–29.9	458	0.157	1.09	(0.97–1.22)	927	0.578	0.98	(0.90–1.06)	0.139
≥30	48	0.025	1.40	(1.04–1.88)	80	0.380	1.11	(0.88–1.38)	0.212
Per 5 kg/m^2^ increase	1388	0.006	1.13	(1.04–1.23)	2920	0.864	0.99	(0.93–1.06)	0.020
*Diabetes status (serum glucose, mg/dL)*									
Normoglycemia (<100)	796		1.00	(Reference)	1640		1.00	(Reference)	
IFG (101–125)	341	0.412	1.05	(0.93–1.20)	693	0.010	1.13	(1.03–1.23)	0.417
Diabetes (≥126 or known diabetes)	251	<0.001	1.36	(1.18–1.57)	587	<0.001	1.83	(1.66–2.01)	0.001
Per 18 mg/dL increase	1388	0.013	1.05	(1.01–1.09)	2920	<0.001	1.12	(1.10–1.15)	0.002
*Alanine aminotransferase (ALT), IU/L*									
<20	481	<0.001	0.79	(0.71–0.90)	299	<0.001	0.35	(0.30–0.40)	<0.001
20–39	691		1.00	(Reference)	1029		1.00	(Reference)	
40–59	139	0.164	1.14	(0.95–1.37)	711	<0.001	3.47	(3.15–3.83)	<0.001
60–79	43	0.056	1.35	(0.99–1.84)	372	<0.001	6.09	(5.39–6.88)	<0.001
≥80	34	0.135	1.30	(0.92–1.84)	509	<0.001	8.63	(7.70–9.66)	<0.001
*Total cholesterol, mg/dL*									
Per 39 mg/dL increase	1388	0.001	0.91	(0.87–0.97)	2920	<0.001	0.55	(0.53–0.57)	<0.001
*Comorbid liver disease*									
Hepatitis B virus infection	14	<0.001	2.71	(1.60–4.59)	287	<0.001	11.80	(10.27–13.57)	<0.001
Hepatitis C virus infection	3	0.369	1.89	(0.47–7.63)	57	<0.001	7.99	(6.11–10.44)	0.047
Liver cirrhosis	4	0.619	1.29	(0.48–3.45)	255	<0.001	19.38	(16.71–22.47)	<0.001
Alcoholic liver disease	15	0.151	1.45	(0.87–2.43)	81	<0.001	1.79	(1.42–2.24)	0.474
*Biliary tract disease*									
Cholelithiasis (gallstone disease)	18	0.004	2.94	(1.40–6.17)	13	0.131	1.53	(0.88–2.64)	0.164
Choledocholithiasis (bile duct stone)	11	<0.001	13.73	(7.58–24.88)	2	0.692	1.32	(0.33–5.30)	0.002

Abbreviations: HR—hazard ratio; 95% CI—95% confidence interval; CCA—cholangiocarcinoma; HCC—hepatocellular carcinoma; IFG—impaired fasting glucose. ^a^ Adjustment for age, sex, smoking, alcohol consumption, physical activity, income, BMI, serum glucose, hepatitis B infection, and choledocholithiasis. ^b^ Adjustment for age, sex, smoking, alcohol consumption, physical activity, income, BMI, serum glucose, liver cirrhosis, hepatitis B virus, and hepatitis C infection.

**Table 3 cancers-14-01709-t003:** Multivariable-adjusted HRs of iCCA and eCCA.

Variable/Group		iCCA Incidence				eCCA Incidence			*p*-Value for Heterogeneity
	No. of Cases	*p*	HR	(95% CI)	No. of Cases	*p*	HR	(95% CI)	
Age, years									
Per 10-year older	821	<0.001	2.33	(1.81–2.99)	567	<0.001	3.03	(2.23–4.11)	0.188
Sex, men (vs. women)	565	<0.001	1.96	(1.64–2.35)	363	<0.001	1.83	(1.47–2.27)	0.628
*Smoking status*									
Never smoker	459		1.00	(Reference)	328		1.00	(Reference)	
Past smoker	82	0.361	1.12	(0.87–1.44)	55	0.257	1.18	(0.88–1.59)	0.791
Current smoker, <1 pack/day	191	0.004	1.33	(1.10–1.61)	119	0.035	1.28	(1.02–1.62)	0.823
≥1 pack/day	57	0.001	1.64	(1.21–2.20)	27	0.419	1.19	(0.78–1.80)	0.219
*Alcohol use, g ethanol/day*									
None	433		1.00	(Reference)	299		1.00	(Reference)	
<10	125	0.145	0.86	(0.69–1.06)	94	0.988	1.00	(0.78–1.27)	0.345
10–39	172	0.065	1.21	(0.99–1.48)	99	0.268	1.15	(0.90–1.48)	0.767
≥40	70	0.006	1.46	(1.11–1.92)	48	0.002	1.65	(1.19–2.28)	0.576
Per 140 g ethanol/week increase	821	0.004	1.10	(1.03–1.18)	567	0.002	1.13	(1.05–1.23)	0.626
*Body mass index, kg/m* * ^2^ *									
<18.5	18	0.126	0.69	(0.43–1.11)	16	0.773	0.93	(0.57–1.52)	0.393
18.5–24.9	504		1.00	(Reference)	326		1.00	(Reference)	
25–29.9	272	0.268	1.09	(0.94–1.26)	183	0.376	1.09	(0.91–1.30)	0.982
≥30	27	0.142	1.34	(0.91–1.98)	20	0.081	1.49	(0.95–2.32)	0.730
Per 5 kg/m^2^ increase	821	0.060	1.12	(1.00–1.25)	567	0.040	1.15	(1.01–1.32)	0.722
*Diabetes status (serum glucose, mg/dL)*									
Normoglycemia (<100)	493		1.00	(Reference)	292		1.00	(Reference)	
IFG (101–125)	192	0.677	0.96	(0.82–1.14)	145	0.070	1.20	(0.98–1.46)	0.099
Diabetes (≥126 or known diabetes)	136	0.054	1.21	(1.00–1.46)	108	<0.001	1.60	(1.28–1.99)	0.059
Per 18 mg/dL increase	821	0.170	1.04	(0.99–1.09)	567	0.026	1.06	(1.01–1.12)	0.476
*Alanine aminotransferase (ALT), IU/L*									
<20	289	0.031	0.84	(0.72–0.98)	192	0.001	0.73	(0.61–0.88)	0.263
20–39	394		1.00	(Reference)	297		1.00	(Reference)	
40–59	88	0.055	1.26	(0.99–1.59)	51	0.912	0.98	(0.73–1.33)	0.206
60–79	28	0.031	1.53	(1.04–2.25)	15	0.682	1.12	(0.66–1.88)	0.340
≥80	22	0.079	1.47	(0.96–2.27)	12	0.800	1.08	(0.60–1.93)	0.397
*Total cholesterol, mg/dL*									
Per 39 mg/dL increase	821	<0.001	0.86	(0.80–0.92)	567	0.872	0.99	(0.91–1.08)	0.010
*Comorbid liver disease*									
Hepatitis B virus infection	9	0.002	2.90	(1.50–5.60)	5	0.049	2.42	(1.00–5.85)	0.747
Hepatitis C virus infection	2	0.668	1.54	(0.21–11.02)	1	0.673	1.53	(0.21–10.89)	0.996
Liver cirrhosis	3	0.416	1.61	(0.51–5.04)	1	0.828	0.80	(0.11–5.75)	0.551
Alcoholic liver disease	7	0.761	1.12	(0.53–2.37)	8	0.059	1.97	(0.98–3.98)	0.283
*Biliary tract disease*									
Cholelithiasis (gallstone disease)	9	0.036	2.87	(1.07–7.66)	8	0.055	3.04	(0.98–9.48)	0.938
Choledocholithiasis (bile duct stone)	5	<0.001	10.90	(4.52–26.28)	5	<0.001	17.54	(7.84–39.26)	0.434

Abbreviations: HR—hazard ratio; 95% CI—95% confidence interval; iCCA—intrahepatic cholangiocarcinoma; eCCA—extrahepatic cholangiocarcinoma; IFG—impaired fasting glucose.

**Table 4 cancers-14-01709-t004:** HRs for CCA and HCC incidence according to liver disorder status.

	CCA Incidence									HCC Incidence								
Variable/Group	Normal Liver Group				Liver Disorder Group				*P_interaction_* ^§^	Normal Liver Group				Liver Disorder Group				*P_interaction_* ^§^
	No. of Cases	*p*	HR	95% CI	No. of Cases	*p*	HR	95% CI		No. of Cases	*p*	HR	95% CI	No. of Cases	*p*	HR	95% CI	
*Alcohol use, g ethanol/day*																		
None	144		1.00	Reference	599		1.00	Reference		357		1.00	Reference	1030		1.00	Reference	
<10	55	0.208	0.89	0.74–1.07	167	0.873	0.97	0.70–1.35	0.638	163	0.823	0.98	0.80–1.19	379	<0.001	0.80	0.70–0.90	0.082
10–39	80	0.077	1.18	0.98–1.42	197	0.513	1.11	0.81–1.51	0.733	150	0.361	0.91	0.73–1.12	482	0.001	0.81	0.72–0.91	0.357
≥40	56	0.063	1.30	0.99–1.71	64	0.005	1.64	1.16–2.31	0.304	54	0.144	1.25	0.93–1.69	230	0.291	0.92	0.79–1.07	0.075
140 g ethanol/week	343	0.027	1.08	1.01–1.16	1045	0.012	1.11	1.02–1.20	0.682	747	0.079	1.07	0.99–1.16	2173	0.875	1.00	0.96–1.04	0.101
*Alcohol use, frequency*																		
None	144		1.00	Reference	599		1.00	Reference		357		1.00	Reference	1030		1.00	Reference	
2/month–2/week	93	0.759	0.98	0.83–1.14	269	0.959	0.99	0.74–1.32	0.916	253	0.649	0.96	0.80–1.15	653	<0.001	0.79	0.71–0.88	0.073
3–4 times/week	99	0.757	1.04	0.81–1.34	168	0.368	1.19	0.82–1.73	0.565	122	0.583	0.92	0.70–1.23	450	0.053	0.86	0.74–1.00	0.655
Almost daily	7	0.003	1.42	1.12–1.80	9	0.018	1.50	1.07–2.11	0.789	15	0.101	1.27	0.95–1.69	40	0.247	0.91	0.78–1.07	0.046

Abbreviations: HR—hazard ratio; 95% CI—95% confidence interval. *P*_interaction_
^§^—*p* value for interaction test between liver disorder status groups.

## Data Availability

Restrictions apply to the availability of these data. The requests to access the dataset from qualified researchers trained in human participant confidentiality protocols may be sent to National Health Insurance Service (NHIS) of Korea at https://nhiss.nhis.or.kr/bd/ab/bdaba021eng.do.

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
