# Peer review of "Comparison of Risk Factors for Cholangiocarcinoma and Hepatocellular Carcinoma: A Prospective Cohort Study in Korean Adults"

_cancers, 2022, doi:10.3390/cancers14071709_

Round 1

Reviewer 1 Report

wish to thank you for giving the opportunity to review this interesting manuscript.
The study presents a cohort of 510,000 Korean patients with a follow-up period of 10 years regarding risk factors for Cholangiocarcinoma and Hepatocellular carcinoma.
The manuscript is well written and the methodology is clearly defined
I have several remarks:
1. Regarding AST/ALT, the normal values are less than 40, what are the basis for selecting the subgroups?
2. Were AST values also assessed?
3. What was the timespan between the laboratory test and the diagnosis of the malignancy. Presumably, in patient with very short time-spans, it is possible that the enzyme elevation represent actual tumor symptom rather than risk factor.
4. Choledochal cysts and cholangitis are among known risk factors for developing CCA.
Did the authors asses these risk factors?
5. The incidence of cholelithiasis seems rather low.
6. How was the prevalence of choledocholithiasis assessed? did the patients had US, if
not then how were they diagnosed? Or is this incidence of symptomatic cholelithiasis/choledocholithiasis?
7. In Table 2 and 3, to improve clarity I suggest adding P-values for each of the parameters in addition to P Heterogeneity.
8. I would recommend elaborating more on limitations in the discussion section: patients with previous cancer were excluded which would make possible selection bias, some known risk factors weren’t assessed, see comment number 3 regarding timing from
test to diagnosis.

Author Response

  1. Regarding AST/ALT, the normal values are less than 40, what are the basis for selecting the subgroups?

Reply: Thank you for your valuable comment. Since ALT and AST levels represent a measure of enzyme activity, their levels vary depending on a number of testing factors and population characteristics. Whatever cutoffs are selected, they are rather arbitrary (Sherman, 1991). An upper normal limit around 40 IU/L has been widely used in Korea and worldwide (Kim, 2014). Additionally, our previous study used the same 40 IU/L cutoffs (Yi et al, 2018). Thus, we selected cutoffs of 40 IU/L for ALT and AST in the current study.

References

  • Sherman KE. Alanine aminotransferase in clinical practice. A review. Arch Intern Med. 1991 Feb;151(2):260-5.
  • Kim SG. Standard Value of Serum Alanine Aminotransferase: Is It Fixed or Varied? Korean J Gastroenterol. 2014;64(4):179-181.
  • Yi, S.W.; Choi, J.S.; Yi, J.J.; Lee, Y.H.; Han, K.J. Risk factors for hepatocellular carcinoma by age, sex, and liver disorder status: A prospective cohort study in Korea. Cancer 2018, 124, 2748-2757.

  1. Were AST values also assessed?

Reply: AST levels were assessed when the participants received a health examination. We have described the baseline AST levels in Table 1.

  1. What was the timespan between the laboratory test and the diagnosis of the malignancy. Presumably, in patient with very short time-spans, it is possible that the enzyme elevation represent actual tumor symptom rather than risk factor.

Reply: Thank you for this valuable comment that reflects an important insight. We investigated the time interval between laboratory tests and the time of the diagnosis of malignant diseases. The median time interval between the health examination and the HCC or CCA diagnosis was 5.3 years (Interquartile range: 2.5-8.1) and 6.1 years (interquartile range: 3.0-8.6), respectively. In total, 315 CCA patients and 852 HCC patients were diagnosed within 3 years of the health examination.

HCC (2068 cases)

CCA (1037 cases)

p

HR

95% CI

p

HR

95% CI

ALT, IU/L

<20

<0.001

0.34

0.29

0.40

0.001

0.79

0.69

0.91

20-39

1.00

1.00

40-59

<0.001

3.63

3.23

4.07

0.417

1.09

0.88

1.35

60-79

<0.001

6.29

5.44

7.27

0.628

0.90

0.59

1.38

≥ 80

<0.001

9.13

7.98

10.44

0.876

0.96

0.61

1.53

An additional analysis was performed after the exclusion of patients diagnosed with cancer within 3 years after the health examination. As a result, the associations of ALT levels with HCC and CCA did not differ significantly from the main analysis results, as shown below.

Indeed, there were many patients diagnosed within 3 years after the health examination, and we agree with your opinion that enzyme elevation may be a manifestation of the tumor. Therefore, we have addressed this issue as one of the limitations in the discussion section.   We also described the median time interval between the health examination and cancer diagnosis in the results section.

  1. Choledochal cysts and cholangitis are among known risk factors for developing CCA. Did the authors asses these risk factors?

Reply: Choledochal cyst, a rare congenital anomaly, was not analyzed in this study. Cholangitis was assessed by the ICD-10 code K83.0 in the claims data, and only 27 subjects were identified. In a further investigation, it was found that CCA occurred in 2 patients with cholangitis (no HCC). The HR for CCA was 20.92 (5.24-83.45, p<0.001) in an age- and sex-adjusted analysis and 4.08 (0.91-18.26, p=0.066) in a multivariable-adjusted analysis.

A characteristic of this study, which is based on claims data, is that infrequent diseases or conditions could not be analyzed. We have described this point as a limitation in the discussion section.

  1. The incidence of cholelithiasis seems rather low.

Reply: In the current study, the prevalence of cholelithiasis in the study population was 0.2%. The prevalence of gallstones detected by ultrasonography among Korean populations who underwent health screenings was reported to be 2.7% in the early 2000s (Chung et al. 2007). Cholelithiasis was assessed based on hospital visits due to cholelithiasis in the current study. Considering that most (around 90%) of cholelithiasis cases are known to be asymptomatic, the prevalence of cholelithiasis in the current study might represent that of symptomatic cholelithiasis.

References

  • Chung YJ, Park YD, Lee HC, Cho HJ, Park KS, Seo EH, et al. Prevalence and risk factors of gallstones in a general health screened population. Korean J Med 2007;72:480–490.

  1. How was the prevalence of choledocholithiasis assessed? did the patients had US, if not then how were they diagnosed? Or is this incidence of symptomatic cholelithiasis/choledocholithiasis?

Reply: Individuals were considered to have choledocholithiasis if they had made at least one visit to a medical institution for choledocholithiasis (K80.3-K80.5) in the interval within 6 months before to 2 months after the baseline health examination date. Considering the wide use of ultrasonography in the diagnosis of gallstones even in the 1990s in Korea (Jung et al, 1992), clinical visits due to cholelithiasis including cholecystolithiasis (K80.0-K80.2) and choledocholithiasis (K80.3-K80.5) are assumed to mainly reflect diagnoses based on ultrasonography. Since people with no symptoms generally do not visit clinics, we assume that symptomatic patients visited clinics and were diagnosed with cholelithiasis/choledocholithiasis.

References

  • HW Jeong et al. Prevalence of gallstones in Korean. Journal of the Korean Academy of Family Medicine 1992;13(7):581-591.

  1. In Table 2 and 3, to improve clarity I suggest adding P-values for each of the parameters in addition to P Heterogeneity.

Reply: Thanks for your comment. In accordance with your recommendation, we have added p-values in Tables 2 and 3.

  1. I would recommend elaborating more on limitations in the discussion section: patients with previous cancer were excluded which would make possible selection bias, some known risk factors weren’t assessed, see comment number 3 regarding timing from test to diagnosis.

Reply: Thank you very much for your advice. We agree with your comments, including the fact that previous cancer patients were excluded, and some infrequent risk factors were not assessed. We revised the manuscript (limitations section) as follows.

“Third, it was difficult to analyze diseases with low incidence or congenital anomalies, so some infrequent risk factors were not assessed. Fourth, the time interval from the health examination to the cancer diagnosis was relatively short for some cases; for example, 315 CCA patients and 852 HCC patients were diagnosed within 3 years after the health examination. Therefore, the impact of some factors might reflect the early presentation of CCA and HCC to some degree. However, excluding the first 3 years of follow-up led to no material changes in the associations compared to the main analysis. Additionally, excluding individuals with previous history of any cancer might have led to bias..

Reviewer 2 Report

This is a 10-year follow-up prospective study conducted in over half a million South Koreans designed to identify risk factors for the development of hepatocellular and cholangiocellular carcinomas and differences among the two types of cancer. The very large number of prospectively enrolled subjects already represents a major strength of the study, independent from some obvious limitations. These include possible genetic and epigenetic mechanisms specific to Koreans and differences in the prevalence of chronic viral infections and dietary habits which may impinge on the generalizability of the data. 

The text requires re-reading to fix some typos.

Author Response

This is a 10-year follow-up prospective study conducted in over half a million South Koreans designed to identify risk factors for the development of hepatocellular and cholangiocellular carcinomas and differences among the two types of cancer. The very large number of prospectively enrolled subjects already represents a major strength of the study, independent from some obvious limitations. These include possible genetic and epigenetic mechanisms specific to Koreans and differences in the prevalence of chronic viral infections and dietary habits which may impinge on the generalizability of the data.

The text requires re-reading to fix some typos.

Reply: Thank you for your comment. As you mentioned, there are some limitations in our study, and other limitations were further commented upon. Additionally, to improve our article’s quality, typos were corrected and English editing by native speakers was performed again.

Reviewer 3 Report

I red with great interest the paper from Cho et al the paper is very interesting, well written and covers a topic of major interest with a large cohot of patients and long follow up

Author Response

I red with great interest the paper from Cho et al the paper is very interesting, well written and covers a topic of major interest with a large cohort of patients and long follow up

Reply:  Thank you for your encouraging comment. We hope that this article will be of interest to the readership of the journal.